# tDCS Anodal Stimulation of the Right Dorsolateral Prefrontal Cortex Improves Creative Performance in Real-World Problem Solving

**DOI:** 10.3390/brainsci13030449

**Published:** 2023-03-06

**Authors:** Jiayue Guo, Jiani Luo, Yi An, Tiansheng Xia

**Affiliations:** School of Art and Design, Guangdong University of Technology, Guangzhou 510090, China

**Keywords:** tDCS, creativity, right DLPFC, design, convergent thinking, divergent thinking

## Abstract

Brain regions associated with creativity is a focal point in research related to the field of cognitive neuroscience. Previous studies have paid more attention to the role of activation of the left dorsolateral prefrontal cortex in creativity tasks, which are mostly abstract conceptual tasks, and less attention to real-world creativity tasks. The right dorsolateral prefrontal cortex is involved in functions such as visuospatial processing, which may have a positive impact on innovative solutions to real-world problems. In this study, tDCS technology was used to explore the effect of anodal stimulation of the right dorsolateral prefrontal cortex on design creativity performance in a real-word problem-solving task related to product design. The experimental task comprised three stages, of which the first two were idea generation stages based on divergent thinking using text and graphics, respectively, whereas the third was the creative evaluation stage based on convergent thinking. Thirty-six design students were recruited to partake in the experiment. They were randomly assigned into anodal stimulation and sham stimulation groups. The results showed that anodal stimulation of the right dorsolateral prefrontal cortex produced a significant positive effect during the creative evaluation stage, promoting the usefulness of ideas (*p* = 0.009); thus, improving product creativity scores. However, there was no significant impact on the idea generation stage (*p* > 0.05), which is dominated by divergent thinking. The results suggest that activating the right dorsolateral prefrontal cortex with tDCS can improve people’s performance in creative activities by promoting convergent thinking rather than divergent thinking. It also provides further evidence that the right hemisphere of the brain has an advantage in solving complex problems that require the participation of visuospatial information.

## 1. Introduction

Creativity is one of the most important thinking skills [1]. There are two forms of creative thinking: divergent and convergent thinking. Divergent thinking is defined as the ability to develop creative ideas by generating multiple solutions; convergent thinking is the ability for the logic of a limited number of solutions [2,3]. After exploring multiple possible solutions to a problem through divergent thinking, convergent thinking can be used to organize their structure and help determine the best one [4,5]. Creative activity includes two processes: idea generation and idea selection [6,7]. Both processes involve divergent and convergent thinking; however, the idea generation stage primarily involves divergent thinking, while the stage of refinement and evaluation of ideas requires convergent thinking [4]. Researchers have developed various creative cognitive tasks to scientifically explore creative thinking in the laboratory, which can be divided into two main categories. One category comprises tasks that mainly rely on idea generation, such as the Alternative Uses Task (AUT). The AUT involves original or unusual ideas for presented stimuli, and is often used to assess divergent thinking. The other relies on idea selection, such as the Remote Associates Test (RAT) and the Compound Remote Associates Test (CRAT). The RAT and CRAT involve integrating seemingly distant concepts or pieces of information to discover or identify new things and is often used to assess convergent thinking. Creativity is a complex structure that requires the participation of various brain functions [8,9]. Studies have focused on the different cognitive processes involved in creativity and their mechanisms [9]. Cognitive operations associated with creativity include analogical reasoning [6], metaphor processing [10], improvisation [11], insight [12], and conceptual expansion [13]. These studies can identify the differential influences of different brain regions in brain networks on various aspects of creative cognition.

In recent years, developments in cognitive neuroscience have played an important role in understanding creativity. Neuroscience research on creative cognition is currently focused on exploring the brain regions that support creative cognition through methods such as electroencephalogram (EEG). The EEG is a record of voltage oscillations caused by internal and external ionic currents in a group of neurons with a certain spatial distribution [14]. The brain’s electrical activity can be examined by quantitative analysis of EEG, such as detecting dementia diseases [14] and cortical activity [15]. By studying the brain network imaging involved in creative cognition through EEG, creativity has made considerable progress in cognitive neuroscience [8]. However, the neuro-electrophysiological and neuroimaging studies are correlation studies; it is often difficult to explain the causal relationship between relevant brain regions and functions. Therefore, researchers have started using transcranial direct current stimulation (tDCS) to explore possible neural mechanisms in the cognitive process of creativity, thereby providing causal evidence of the roles of specific brain regions in each process. tDCS provides a non-invasive method to help establish a causal relationship between a specific region of the brain and its underlying functions [16]. Several researchers have used tDCS to observe the mechanism of the dorsolateral prefrontal cortex (DLPFC) in creative cognition. DLPFC is related to functions such as cognitive control and working memory and facilitates the integration of semantically distant concepts [6], creative idea selection [17], reflective decision-making performance [18], response inhibition [19], and other processes.

A previous study applied tDCS to the left/right DLPFC (the reference electrode was located in the contralateral supraorbital area) to study its effect on creative cognition; only anodal stimulation of the left DLPFC improved RAT performance, whereas that of the right DLPFC did not [20]. Researchers believe that the left dorsolateral prefrontal region contributes to creative and logical complex verbal problem solving [20]. Compared with sham stimulation, when tDCS stimulation was applied to the bilateral DLPFC, the stimulation type of L+R− (left anodal and right cathodal) significantly improved the scores of the three dimensions of the AUT task (fluency, flexibility, and originality), while L−R+ had no significant effect [21]. In a study on the effects of tDCS stimulation of the anode DLPFC on intuition and analytical thinking, the anode tDCS of the right DLPFC was found to improve analytical judgment and decision making [18]. In addition, researchers found that both anode and cathode tDCS stimulated in the right DLPFC can improve response inhibition [19]. Comparing these studies shows that the researchers used different creativity tasks and obtained different results. 

Most existing studies using tDCS to explore the DLPFC and creativity use language-related tasks, such as AUT and RAT, which usually proves that the positive stimulation of the left DLPFC can promote creative cognition, and the left DLPFC facilitates idea generation and selection. In contrast, the role of the right DLPFC is often not significant [20]. Due to the experimental design and subject constraints, AUT and RAT tend to sacrifice some ecological validity [22], and the correlation with real-world creativity varies widely [20]. Unlike results using only a verbal task, anodal stimulation of the left and right DLPFC with tDCS has been shown to increase creativity using a visualization AUT task [23]. These studies suggest that the nature of the task (verbal or visuospatial) may influence the effectiveness of DLPFC stimulation in tDCS tasks. These tasks seem difficult to accomplish with words alone. Therefore, we suggest that tDCS stimulation of the right DLPFC may have a more pronounced effect on complex creative cognitive tasks such as real-world problem solving. Because the experiment aimed to study the role of brain regions in different creative cognition processes, the design process was simplified into three stages in this study: word conception and prototype generation for problem definition and prototype idea divergence, and a final proposal stage requiring idea evaluation and refinement. Considering the important role of the DLPFC in divergent and convergent thinking, we hypothesized that activating the right DLPFC could effectively promote participants’ creativity performance. As the right DLPFC is associated with spatial/visual cognition, we hypothesized that its anodal stimulation may have an effect during the prototype generation phase (divergence through graphical thinking) and the final proposal phase (drawing and refining the product), but less so during the textual conception phase. Previous studies have shown that the DLPFC is the main area of the brain’s control executive network and is related to cognitive inhibition and working memory. We predicted that activating the right DLPFC may reduce the number of prototype sketches during the prototype generation phase, whereas activating the right DLPFC may enhance the usefulness of the results during the later stages of idea evaluation and refinement. 

The paper is organized as follows. The second section describes the materials and methods used in our study. In the third section, we present the analyzed results of the experiments. Then, the results and findings are discussed. The last section summarizes the conclusions.

## 2. Materials and Methods

In this study, an ecologically effective design creativity task was used to investigate the effect of anode tDCS stimulation of the right DLPFC on creativity. A double-blind design was used to divide participants into a sham stimulus group and an active stimulus group, and creativity performance was measured by several dimensions (including text fluency, prototype fluency, usefulness, novelty, and creativity) of idea generation and idea selection during the design creativity task.

### 2.1. Participants

To meet the basic requirements for professionalism in experimental tasks, we recruited 36 participants, including undergraduate and graduate students majoring in industrial design and product design from the Guangdong University of Technology (including 13 males and 23 females; average age *M* = 21.64, *SD* = 2.35). The participates were randomly assigned into two groups (18 people per group). According to the sample size calculation based on power analysis using G*Power version 3.1.9.7, in similar tDCS experiments [24], input-related parameters (effect size dz = 0.93, α error probability *p* = 0.05, power = 0.8) found that a sample size of 12 is sufficient to obtain reliable results. All participants were right-handed, native Chinese speakers with normal or corrected vision [25]. They had completed at least one year of professional design study with a certain basis for product hand painting. As academic and labor experiments (e.g., design training [26]) may affect participants’ creative performance, we recorded the received education years of graduate and undergraduate students.

The participants were required to fill in the report before the experiment and were determined to be healthy individuals with no history of neurological or psychiatric disorders, brain surgery, or intracranial metal implantation. The experiment required the participants to not consume any alcohol or caffeine within 24 h before the test, and to maintain a good mental state to avoid the influence of sleep deprivation on the experiment’s results. All experiments in this study were reviewed by the Ethics Committee of Guangdong University of Technology, and written informed consent was obtained from each participant. After the experiment was over, participants were provided with CNY 40 as a reward for participating in the experiment. 

### 2.2. Materials

#### 2.2.1. tDCS

The study used a double-blind trial. All participants, experimenters, and raters were blind to the intervention type. Participants received one of the two anonymous codes randomly generated by the computer, which corresponds to the anodal stimulation group (N = 18) and sham stimulation group (N = 18), respectively. tDCS was performed using STARSTIM 8 equipment (Neuroelectrics, Barcelona, Spain). The procedures for the two groups were named anonymously by these two codes; therefore, both participants and experimenters were blind to the type of intervention. A pair of rubber electrodes (5 × 5 cm^2^) covered with a saline-soaked sponge was used, and the current level was set to 1.5 mA for safety reasons [21,27]. According to the 10−20 International EEG Electrode Placement System, the stimulating electrode was placed at F4 (right DLPFC) [28,29], and the reference electrode was placed on FP1 (contralateral orbitofrontal area) [30,31,32,33], as shown in Figure 1. Participants in the anodal stimulation group received 20 min of tDCS, in which the current was steady at 1.5 mA, rising and falling for 30 s each. The positioning of electrodes in the sham stimulation group was the same as that in the stimulation group. The sham group also experienced 30 s with a 1.5 mA current of the rise and fall at the beginning and end of the program, respectively; all participants reported feeling the current. Studies have shown that this can mimic the sensations produced by real stimuli without producing significant neuromodulatory effects in the cortex [34]. The electrodes remain in the position for the rest of the time without current to achieve the blinding effect.

#### 2.2.2. Design Tasks

Referring to the previous design process [35], we divided the design task into three stages: the word concept stage, which uses natural language divergence to define the problem; the prototype generation stage, which requires sketches to express different conceptual ideas; the final proposal stage, which requires idea evaluation and refinement. The first two phases were mainly concerned with idea generation, whereas the third phase was concerned with idea selection. According to the criteria for a creativity task suitable for empathy assessment [36], a suitable task must meet three requirements: first, the results or responses of the task must be observed; second, the task results should be open enough to ensure flexibility and innovation; third, the research participants should not have individual differences in the task baseline performance that are too large. Therefore, we chose “containers that bring food to school” as the theme of this design task [37,38]. This theme is related to students’ daily lives and is a relatively common concept. To a certain extent, this can avoid the influence of individual knowledge reserves and experience deviations in the design results. Simultaneously, to avoid cognitive fixation, the participants were limited to the subject through text, without using specific examples or images. In the text conception stage, the participants needed to use their native language to write out their ideas about the topic and record them on A4 paper with grids. Thereafter, they were required to generate a prototype of the theme in the form of a hand-drawn expression, which was also recorded on A4 paper with grids. Finally, the participants were required to select the most creative plan from the first two stages, refine it manually, and produce the final product plan with the necessary text descriptions.

#### 2.2.3. tDCS Adverse Effects Questionnaire

To ensure the safety of the experiment, the participants were notified that they could ask to terminate the experiment any time without bearing any consequences should they encounter any unbearable feelings during the stimulation process. Additionally, double-blind experiments involving sham stimuli required participants to be unable to determine whether they were receiving an electrical stimulus [39]. The effectiveness of the sham-blinding method can be measured by comparing the frequency and intensity of cutaneous sensation when receiving the anode stimulus or sham stimulus [39]. To determine the blinding effect and ensure the safety of the experiment, the participants were also asked to guess whether they had received the stimulation and fill out a 4−point Likert scale after the stimulation procedure, through which the side effects of the stimulation were evaluated. The content included some of the most common side effects of tDCS, such as itching, tingling, headaches, and fatigue [40,41].

### 2.3. Experimental Procedure

Because the effect of tDCS can last for a long time after the end of electrical stimulation [42], the experiment adopted an offline mode to avoid interference with the participants from wearing the device. Our experiment had a between-subject design; all participants experienced the same experimental procedure, as shown in Figure 2. The variable in our experiment is a type of stimulus (anodal stimulation or sham stimulation). After receiving a 20 min tDCS (or sham stimulation), the participants were asked to perform a 40 min product design task. The product design task included three stages. In the first stage, the participants were asked to conceive the design theme within five minutes using their native language. To facilitate statistics in the experimental environment, the participants were provided with some A4 papers with multiple 6.5 × 4.0 cm grids printed on each paper. They were directed to use their creativity through the experimental instructions instead of limiting their thinking or self-censoring. Further, they had to write down as many thoughts as they could within the allotted time, and record each different thought in a new box. After the word conception stage, the participants were provided with a new A4 paper and asked to conduct a 15 min prototype divergence on the topic. This step mainly employed visualization and required the participants to be creative, and to let go of the limitations of thinking. Following these directions, they had to propose as many different prototype solutions as possible. Finally, the participants had 20 min to evaluate the design ideas generated in the first two stages, select the idea that they thought was the most creative, finalize it on an A4 paper, map out the main functional components in words, and write brief design and operation instructions. 

### 2.4. Measures

#### 2.4.1. Creativity Scoring

Previous research has suggested that qualitative research on creativity should be conducted in a domain-specific manner [36]. Standard methods for evaluating early-stage idea generation in product design are currently lacking. Therefore, we refer to the Torrance test. Four indicators, namely fluency, originality, elaboration, and flexibility are typically used to evaluate the number of ideas, novelty of concepts, level of detail, and variety of ideas. Since this experiment divided the design process into different stages, in the idea generation stage in the early design stage (including text divergence and sketch divergence), the main requirement is that the participants generate as many ideas as possible through divergent thinking, and therefore, originality and elaboration were not considered. As for flexibility, regarding the diversity of testing ideas, since the divergence of designers’ thinking in the early stage is usually uncertain and vague, it is difficult to clearly distinguish categories. Therefore, this indicator was also excluded. Consequently, fluency is used as an objective indicator for assessing creativity in the early stages of design. Fluency refers to the number of ideas generated within a given period, and creative people are generally considered to generate more ideas [43].

At the design output stage, the final solution provided by the participants was evaluated using the Moss metric, which is mainly used to evaluate creative products and is still widely used [37,44]. The Moss metric calculates the creativity of a product by using a combination of two factors: novelty and usefulness. Novelty is the basis of creativity; however, novelty without usefulness will introduce unavailing products, such as a safe made of soap bubbles. Therefore, when evaluating creativity, the two indicators of novelty and usefulness should be considered simultaneously [45]. Usefulness is determined by comparing the degree to which the functional requirements of the product meet the standard or instructor’s solution. Standard solutions refer to those that meet basic functions while ensuring product quality. Depending on how successful the solution is at the functional level, the possible values vary from 0 to 3 as follows: 0, does not meet the basic functions; 1, only completes the basic functions; 2, reaches the level of quality of the instructor’s solution; 3, the solution is functionally superior to the standard solution. Novelty is determined by the inverse probability that such a solution occurs in a homogeneous set of solutions. Solutions were also scored between 0 and 3 according to the frequency of concepts: 0, very common solution (>10% of similar concepts); 1, common solution ([10%, 5%]); 2, uncommon solutions (<5%); 3, very rare or original concepts (1%). Finally, the degree of creativity of the design proposal was calculated by multiplying the novelty and usefulness scores. In this measurement mode, the creativity score of the final solution was between 0 and 9. According to Runco’s review of creativity, creative products must be original and practical (both novel and useful), which is also the consensus standard for truly creative products in creativity research [45]. In the Moss metric system, regardless of whether it lacks novelty or does not meet the function requirements, the result is “no creativity,” which is consistent with the consensus in the field of creativity mentioned above.

Using empathy evaluation technology in accordance with the aforementioned evaluation methods, two Doctors of Industrial Design and a third-year postgraduate student (two females and one male) were invited to evaluate the results, independently. The raters were required to read all the solutions and obtain a general understanding of the overall level before scoring, to ensure that the evaluation was based on the relative standards of all the tested solutions, rather than absolute standards in the professional field. Simultaneously, to ensure the consistency of the judges’ subjective standards for a specific dimension and to ensure the separation between different dimensions, they were required to evaluate all proposals on one dimension, such as novelty, before evaluating the next dimension, such as usefulness [36]. All the raters were blind to the intervention type.

#### 2.4.2. Statistical Analysis

All statistical analyses were performed using SPSS version 25 (IBM, Armonk, NY, USA). Analysis of variance (ANOVA) was used to test the significant difference between the creativity of the experimental and control groups and its sub-variables and the basic information (including ages, years of education, gender, number of hours slept, number of stimulants taken, and Edinburgh handedness inventory). When *p* < 0.05, it indicated that there were significant differences between the variables. For participant-reported adverse reactions, Kruskal-Wallis calculations were used to examine differences between the groups [41].

## 3. Results

After receiving tDCS stimulation (anodal or sham stimulation), the participants completed the product design plan for the design theme according to the design process. The output example is shown in Figure 3.

### 3.1. Results of the Basic Information

The basic information reported by the participants is shown in Table 1. Analysis revealed no significant differences between the variables (*p* > 0.05). Moreover, the participates cannot recognize the difference during the sham stimulation and anodal stimulation.

### 3.2. Results of the tDCS Adverse Effects Questionnaire

Adverse effects reported by the participants are shown in Table 2. For each adverse effect, a Kruskal-Wallis test was performed to check for group differences. Analysis revealed no significant differences between the groups regardless of the type of adverse effect, indicating no noxious effects of stimulation and a successful sham arrangement.

### 3.3. Creativity Scoring

The results of the univariate ANOVA are shown in Table 3. There was no significant difference between text fluency (*F*(1, 34) = 0.19, *p* = 0.663, *ŋ_p_*_2_ = 0.006), and prototype fluency (*F*(1, 34) = 1.39, *p* = 0.247, *ŋ_p_*_2_ = 0.039) for the participants in the anode stimulation group and the sham stimulation group at the stage of using natural language for word conception as well as the stage of using graphic thinking for prototype generation. This indicated that activation of the right DLPFC had no significant effect on thinking fluency. In the final proposal output stage, experts evaluated the usefulness and novelty of the final program with consensus, and the results showed that the usefulness score (*M* = 1.56, *SD* = 0.49) of the works in the anodal stimulation group was significantly higher than in the sham stimulation group (*M* = 1.06, *SD* = 0.60), (*F*(1, 34) = 7.63, *p* = 0.009, *ŋ_p_*_2_ = 0.183), but had no significant impact on the novelty of the work, (*F*(1, 34) = 0.03, *p* = 0.870, *ŋ_p_*_2_ = 0.001). According to the Moss measure, the final creativity of the scheme was obtained by multiplying usefulness and novelty, and the results showed that the creativity of the final product design of the anodal stimulation group was significantly higher than that of the sham stimulation group (*F* (1, 34) = 5.70, *p* = 0.023, *ŋ_p_*_2_ = 0.144).

## 4. Discussion

This study had two main purposes: first, to verify the relationship between the influence of the right DLPFC on creativity; second, to explore the possibility of influencing real-world creativity performance through tDCS stimulation.

### 4.1. Relationship between the Influence of the Right DLPFC on Creativity

Few recent studies on the relationship between tDCS and creativity consistently found that the left DLPFC is connected to creative performance related to language reasoning, which proves the important role of the left DLPFC in the creative thinking process during language reasoning [21,24,46,47,48]. However, there is a lack of research on the role of the right DLPFC in creative thinking. Previous research found that such anodal-excitation effects were not observed in all tDCS studies, and their effects may be related to the experimental task [49]. Previous studies found that the anodal-excitation effects were more obvious in research related to the motor area. Whereas, in most cases related to cognitive research, only the anodal stimulation of the target cortical area caused an increase in function. Adversely, cathodal stimulation in the same area was weak or absent. This may be due to cognitive function often being supported by rich brain networks, reflecting the existence of compensatory processes. While focusing on the right DLPFC, we found that studies reporting significant effects of tDCS on the right DLPFC tended to use more ecological and realistic experimental settings or tasks. In stressful situations, anodal stimulation of the right DLPFC can alleviate stress-induced creative blockage [50]. In an improvisation task, anodal stimulation of the right DLPFC was shown to promote the creative performance of novices [51].

When the problem is familiar and well structured, the left prefrontal cortex is usually activated [52]. In contrast, when faced with an unfamiliar or poorly structured problem, the right DLPFC is usually activated; the former may be dominated by verbal reasoning, while the latter involves spatial/visual processing. However, our experimental results show that in ill-structured real-world problem-solving tasks requiring visual/spatial information processing, such as product design, tDCS stimulation of the right DLPFC significantly improves the creativity of the final solution. Compared to the simple text conception stage during the early stage of ideation, the content generated in the specific task stage indicates that the stimulation effect of the right DLPFC becomes significant only in the final design completion stage, which is consistent with previous research [52]. Therefore, we believe that the activation of the right DLPFC can promote better creative performance when solving real-life problems. This result may be related to its role in cognitive control and working memory.

Regarding the specific stage of creativity generation, a study reported the activation of brain regions in the stage of idea generation and evaluation in the book cover design task, according to the results of fMRI [53]. The results of our study confirm this finding. During the design task phase involving two processes, anodal stimulation of the right DLPFC by tDCS had no significant effect on the results of tasks (word ideation, prototype generation) in the idea generation phase. In contrast, in the final proposal phase, anodal stimulation of tDCS significantly increased the usefulness of the solution. Since novelty and usefulness are inherent properties of creative products, an increase in the latter significantly increases the creativity of solutions without compromising novelty. The differences between current research and other similar works are shown in Table 4.

### 4.2. Possibility of Influencing Real-World Creativity Performance through tDCS Stimulation

In experiments on the relationship between the left prefrontal cortex and creativity, anodal stimulation increased the appropriateness of participants’ responses and decreased novelty, which they believe may indicate the impact of cognitive control on creativity [27]. A similar trend was observed in our experiments. In the experiment, anodal stimulation of the right DLPFC slightly reduced the novelty of the final outcome, and even reduced the number of prior thoughts generated, although neither was significant.

Concerning the different thinking aspects of creative cognition, in our experimental task, the early stage of idea generation mainly involves divergent thinking, while the later process of evaluating and refining ideas requires the participation of more convergent thinking, and the result can receive feedback through user satisfaction with product functions. Specifically, after the designer creates many possible solutions through divergent thinking in the prototype generation stage, in the next stage, the designer uses convergent thinking to select the best possible solution that meets the solution criteria and refines it. This process involves analytical thinking and focused attention [5]. Thus, our findings suggest that in creative thinking, the right DLPFC plays a greater role in convergent than in divergent thinking. A more ecologically valid way of assessing people’s creativity than a simple divergent thinking test in the laboratory is to ask them about their real-life creative achievements [54].

### 4.3. Contributions and Limitations

The reasons for creative achievements, in reality, are often very complex. They are not only related to creative thinking but also to personality, opportunities, resources, and environment. These variables interfere with the accuracy of creativity evaluation, making it difficult to establish a causal relationship between cognitive nerves and creative thinking. Therefore, our experiments combined tDCS with a simplified, well-flowed, practical design task to differentiate creative thinking modes through design phase transitions. It not only ensures the ecological validity of creativity but also avoids the interference of irrelevant variables to a certain extent so that it can better explore the role of specific brain regions in creative activities in reality. Simultaneously, we must admit that creative thinking in the real world is a complex process in which divergent and convergent thinking are often not discrete but intertwined and complementary.

In our experiment, we used a cross-sectional study without long-term neuromodulation. Future research is suggested to design a study that involves multiple sessions of tDCS to allow for long-term neuromodulation and its impact on the mental health of participants, and at the same time, ethical issues should be considered. Additionally, it would be important to evaluate the extent to which the improvements or deteriorations found are sustained over time. Moreover, some researchers believe that in the creative design process, there is more divergent thinking in the prototype generation stage [4]. Still, there may be both divergent and convergent thinking in other stages. Additionally, in this study, only fluency was assessed for the results of the text and prototype stages of the experimental task, which may not fully reflect the creative thinking of the participants at this stage. In future experiments, other methods can be considered to conduct standardized coding analysis on this part of the content and to further study other attributes of the previous ideas, except for fluency.

Since this task requires participants to have received training in design thinking and processes and includes professional requirements for hand-painting skills, we recruited students only majoring in design. It provides a certain reference for the application of creativity research in the field of real creativity. Future study is suggested to increase the sample size and evaluate a standardized population, as this would provide a more generalizable understanding of the results. Additionally, the selected sample is very young, with an average age of 21 years, and the results could vary greatly with an older population. Previous studies have shown that old people and young people have different creativity performance, possibly because older people have more experience and knowledge [55]. Therefore, it would be valuable to consider the effects of age on the outcomes of the study and discuss the implications of these results for future research. Finally, it would be worth considering the control of variables, such as state, as the participants in our study are healthy and have a balanced brain activity without pathology in their mood states. However, the results of mood effect for tDCS on the DLPFC are still complex [56]. This aspect of the study could benefit from further exploration and analysis.

## 5. Conclusions

This research shows that activating the right DLPFC with tDCS can improve an individual’s performance in creative activities by promoting convergent thinking rather than divergent thinking. Additionally, it provides evidence that the right hemisphere of the brain has an advantage in solving complex problems that require the participation of visuospatial information. The positive effect of tDCS on the right DLPFC is not obvious in the early design stage when divergent thinking is primarily used; but it plays a significant role in the plan refinement stage, which requires the participation of convergent thinking. However, the ability of tDCS to affect creativity in the right DLPFC brain area is related to the nature of the task. Compared to other studies using verbal and non-verbal experimental tasks, anodal stimulation of the right DLPFC showed a more pronounced effect in design tasks requiring visuospatial thinking. We hope these conclusions provide a valuable theory for further research on the function of the right DLPFC and creativity based on tDCS and provide a reference for the development and application of tDCS technology in the future.

## Figures and Tables

**Figure 1 brainsci-13-00449-f001:**
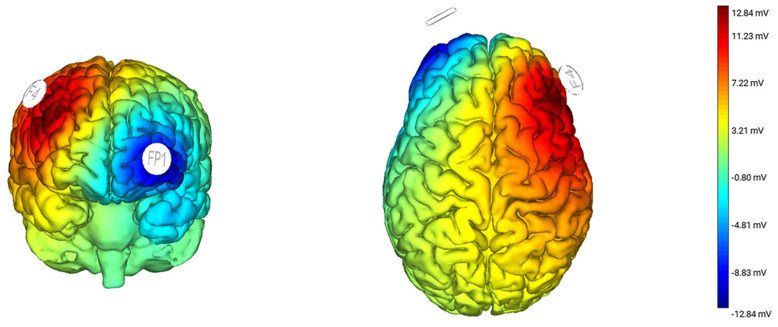
Electrode configuration and current flow in the cathodal tDCS group. Pictures are executed using the software Neuroelectrics^®^ Instrument Controller (NIC, version 2.1.0).

**Figure 2 brainsci-13-00449-f002:**
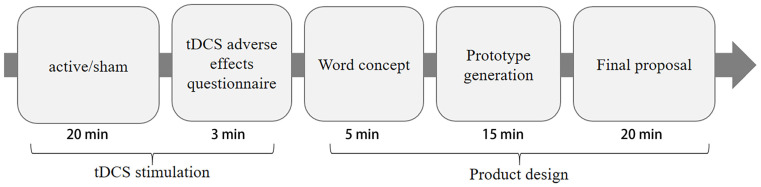
Experimental procedure.

**Figure 3 brainsci-13-00449-f003:**
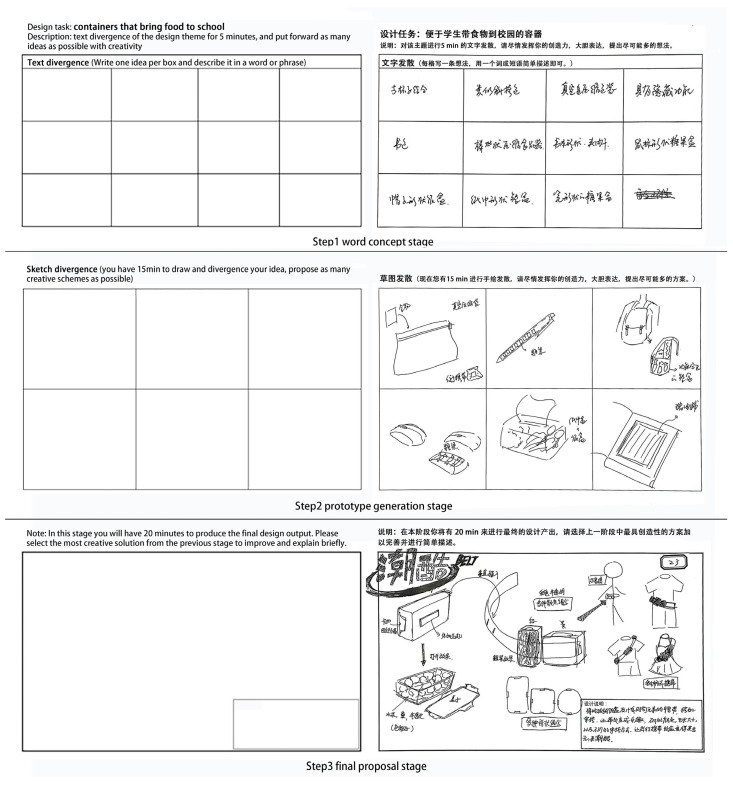
Design method and design example.

**Table 1 brainsci-13-00449-t001:** Results of the basic information.

	Anodal tDCS Group (*n* = 18)		Sham Group (*n* = 18)		*p*
	M (95%CI)	SD	M (95%CI)	SD	
Age (years)	21.22 (20.39 to 22.05)	1.66	21.61 (20.69 to 22.53)	1.85	0.51
Years of education (years)	13.56 (13.30 to 13.81)	0.51	13.61 (13.36 to 13.86)	0.50	0.74
Gender: n (%) Males	7 (38.90%)		6 (33.33%)		0.73
Number of hours slept	6.90 (6.37 to 7.43)	1.07	6.74 (6.30 to 7.19)	0.89	0.64
Number of stimulants taken	1.28 (0.69 to 1.86)	1.18	1.17 (0.59 to 1.74)	1.15	0.78
Edinburgh handedness inventory	81.31 (72.20 to 90.07)	17.97	85.81 (75.81 to 95.81)	20.11	0.47

Note: M = mean; SD = standard deviation; CI = confidence interval.

**Table 2 brainsci-13-00449-t002:** Adverse effects of tDCS. Data are rank mean.

Adverse Effect	Sham	Anodal	Kruskal-Wallis	*p*-Value
Scalp pain	8.60	8.33	0.026	0.873
Tingling	10.25	12.00	0.750	0.386
Itching	14.71	11.42	2.053	0.152
Burning sensation	4.67	4.00	0.333	0.564
Sleepiness	6.25	5.70	0.100	0.752

**Table 3 brainsci-13-00449-t003:** The scores of the sham stimulation group and the anodal stimulation group.

	Sham Stimulation Group		Anode Stimulation Group		*F*
	*M*	*SD*	*M*	*SD*	
Text fluency	13.28	4.35	12.67	3.99	0.19
Prototype fluency	10.28	4.86	8.72	2.78	1.39
Usefulness	1.06	0.60	1.56	0.49	7.63 **
Novelty	1.48	1.06	1.43	0.96	0.03
Creativity	1.09	0.99	2.15	1.59	5.70 *

Note: * means *p* < 0.05, ** means *p* < 0.01. *M* = mean; *SD* = standard deviation; *F* = F-test value

**Table 4 brainsci-13-00449-t004:** The differences between current research and other similar works.

Adverse Effect	Stimulation Type	DLPFC Region	Experimental Task	Significant Result
Xiang, 2021 [23]	Anode/cathode	Left	CRAT/AUT	Divergent thinking
Koizumi, 2020 [26]	Anode	Left	AUT	No significant
Zmigrod, 2015 [52]	Anode	Left	CRAT	Convergent thinking
Huang, 2022 [53]	Anode/cathode	Left	riddle task	Novelty

Note: CRAT = compound remote associates test; AUT = alternative uses task.

## Data Availability

The datasets generated during and/or analyzed during the current study are available from the corresponding author on reasonable request.

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
