# Peer review of "tDCS Anodal Stimulation of the Right Dorsolateral Prefrontal Cortex Improves Creative Performance in Real-World Problem Solving"

_brainsci, 2023, doi:10.3390/brainsci13030449_

Round 1
Reviewer 1 Report
- Introduction section is too long.
- There are a lot of unnecessary and really old references, such as 68, 65, 36, 33, 5. Some might been classics, but this is cutting-edge journal, not educational institution. However, novel references are missing.
- Comment doi: 10.1016/j.brs.2018.12.003
https://doi.org/10.1016/j.ijpsycho.2021.01.014
https://doi.org/10.1038/s41598-019-43626-4
- Did you discard auto-suggestion in the student population?
- Line 241: What is the influence of the sex distribution to the conclusions? Do man and woman have same creative capabilities?
- Lines 251-254: Did you prove that by some test?
- Lines 261-2: Explain what do you mean with "double blind"? How did you execute it?
- Lines 305-7: Questionable. Explain further.
- Line 315: Provide more details about sham stimulation.
- Figure 2: Are the members of active/sham groups the same in all experiments or they are mixed?
- Line 356: "accepted today [60,68]. " 68 is from 1966?
Reviewer 2 Report
The authors present the paper entitled “tDCS anodal stimulation of the right dorsolateral prefrontal cortex improves creative performance in real-world problem-solving”
This study uses tDCS technology to explore the effect of anodal stimulation of the right dorsolateral prefrontal cortex on design creativity performance in a real-world problem-solving task related to product design.
The article presents the following concerns:
- Please, at the end of the introduction, including the structure of the manuscript.
- Include a quantitative value in the abstract to highlight the findings.
- Lines 87 to 105: Authors must validate if these lines correspond to the introduction or methods section.
- Authors must define convergent and divergent thinking in the introduction.
- It is advisable to synthesize the introduction into a maximum of 2 pages to improve understanding.
- It is recommended to add a brief introduction between section 2 and subsection 2.1.
- Avoid using pronouns; instead, use the passive voice.
- In line 241: Please add information about the ages of the rest of the students who attended the study since only an average of 13 of them is mentioned.
- The authors must describe and analyze the possible effects of the academic and labor experience between undergraduate and graduate students.
- It is recommended to add a brief introduction between sections and subsections.
- Please fix table 1.
- The authors must update the references section since less than 25 percent of the citations are under 2017.
- Line 66 can be justified with recently advanced in EEG measurements, I recommend you update by considering these: Impact of eeg parameters detecting dementia diseases: a systematic review; Cortical activity at baseline and during light stimulation in patients with strabismus and amblyopia.
- A comparative Table between the results of this research and other similar works must be added.
- Add hyperlinks to tables, figures, and references.
- Apostrophes must be avoided.
- According to Turnitin, authors must decrease duplicity from 40% below to 20%.
The following misspelling should be checked:
- The word “usefulness” appears repeatedly in this text. Consider using a synonym in its place: “help”, “service”, “use” or “benefit”
- line 91: “ In contrary…” should be rewritten as “On the contrary…”
- line 214: The to-infinitive “to generate” has been split by the modifier “continuosly”. Avoiding split infinitives can help your writing sound more formal like “to generate ideas and select evaluations continuously”
- line 437: The phrase “it was more” may be wordy. Consider changing by “more than”
- line 460: The phrase “in terms of” may be wordy. Consider changing by “Regarding”
- line 500: “generation stage [6], but there may…” A knowledgeable audience might find this sentence hard to read. Consider breaking it into two: “generation stage [6]. Still, there…”
Reviewer 3 Report
Firstly, I am writing to express my gratitude for the opportunity to review the research article “tDCS anodal stimulation of the right dorsolateral prefrontal cortex improves creative performance in real-world problem-solving”. I am honored to have been selected to contribute to the peer-review process for Brain Sciences.
I understand the critical importance of rigorous evaluation in academic research and am eager to lend my expertise to this process. I am confident that my analysis will be of value to the authors and help ensure that the work is of the highest quality.
Thank you for entrusting me with this important task. I look forward to the opportunity to provide a thorough and constructive review.
In summary, the article focuses on the research of the brain regions associated with creativity in the field of cognitive neuroscience. Previous studies have mostly focused on the left dorsolateral prefrontal cortex in abstract tasks, but this study looks at the right dorsolateral prefrontal cortex in real-world problem-solving tasks. The study used tDCS technology to explore the effect of stimulation on the right dorsolateral prefrontal cortex on design creativity in a product design task. 36 design students participated in the experiment, which was divided into anodal stimulation and sham stimulation groups. The results showed that stimulation of the right dorsolateral prefrontal cortex had a significant positive impact on the creative evaluation stage, but not on the idea generation stage. The results suggest that stimulating the right dorsolateral prefrontal cortex can improve creative performance by promoting convergent thinking and provide further evidence for the role of the right hemisphere in solving complex problems.
I would like to make a series of improvement suggestions to the authors:
-
A suggestion for improvement would be to explain the reason for using only design students as the sample population. It might be more interesting to evaluate a standardized population, as this would provide a more generalizable understanding of the results.
-
One suggestion for improvement is to address the impact of age on the development of creativity. The selected sample is very young, with an average age of 21 years, and the results could vary greatly with an older population. It would be valuable to consider the effects of age on the outcomes of the study and to discuss the implications of these results for future research.
-
Provide a more comprehensive description of the participants, including any demographic information that may be relevant to the study. This will help the reader understand who was included in the study and how representative they are of the population being studied.
-
Why was the decision made to use 1.5 mA instead of 2 mA? We know that intensity is related to the improvement of neurocognitive effects, so it seems more reasonable to use 2 mA. Could you provide some clarification on this choice?
-
It would be worth considering the control of variables such as state, as we are dealing with healthy subjects who will have a balanced brain activity without pathology in their mood states. That being said, I wonder how anodal tDCS on the r-DLPFC will affect the mood of the participants. This aspect of the study could benefit from further exploration and analysis.
-
A suggestion for future research would be to design a study that involves multiple sessions of tDCS to allow for long-term neuromodulation and its impact on the mental health of participants. Additionally, it would be important to evaluate the extent to which the improvements or deteriorations found are sustained over time
The rest of the research article appears to be well written and in compliance with academic standards. The methodology, results, and discussion are all presented in a clear and concise manner, making it easy for readers to follow the study's progression. The use of appropriate statistical tests and the thorough interpretation of the results demonstrate a strong understanding of the research topic. Overall, the research article provides valuable insight and contributions to the field
I want to express my sincere appreciation for the time and effort that you have invested in your research. Your dedication to advancing knowledge in your field is truly admirable
Keep up the good work, your dedication and hard work is evident in the quality of your research.
It has been a pleasure reviewing your work and I am confident that with the suggested revisions, your paper will make a valuable contribution to the field. I wish you all the best in your continued research and look forward to seeing your future publications.
Best regards,
Round 2
Reviewer 1 Report
- At the end of the Introduction, I meant that you describe what is the organization of the paper like e.g. "The paper is organized as follows. The 2nd section describes.... The last section summarizes the conclusions."
- Lines 135-137 missing citation of the reference https://doi.org/10.3389/fpsyg.2023.1052257. As you can see, from Ithenticate report, this sentences are from that source.
- In explanation of "double blind" I'm interested also in: are they able to feel the stimulation if it is sham, not real, and can they recognize the difference?
- Can you include wave-form of the stimulation? Also, for the sham stimulation.
- The paper should be rewritten to reduce similarity, which is now 36%.
- Do you consider this research ethical? Your subjects are humans. The stimulation influences their brains, and perhaps personalities? Do you have some researches that proves that they are going to be OK? Changing the creativity is, in fact, influence to their personality? Is it only small temporal change or permanent change in human personality (if someone is not creative, you can make it to be creative)? This is also an interesting question in sense of AI science.
Reviewer 2 Report
The Manuscript can be accepted
Author Response
Thank you for effort reviewing our paper and comments.